# Applicability Analysis of Trunk Drainage Sewer System for Reduction of Inundation in Urban Dense Areas

## Changjae Kwak

National Disaster Management Institute, Ulsan 44538, Korea; water203@korea.kr

**Abstract:** Urban development naturally aggravates flood damage, causing severe damage yearly. Preparation for flood damage is a part of urban planning, but it is not easy to establish clear mitigation measures in densely populated urban areas. This study analyzed the applicability of trunk drainage sewers as an alternative to installing abatement facilities, a typical structural measure for reducing flood damage in dense urban areas. The study areas included three areas in South Korea where flood damage had previously occurred, and the input parameters of the flood analysis model were calibrated based on the measured runoff, followed by testing with inundation traces. The results of three watersheds were qualitatively evaluated using the Lee Sallee Shape Index (LSSI) method. The applicability of the trunk drainage sewer system in the Gunja and Dowon watersheds were "Excellent" and "Good" in the Dorim watershed. The analysis results for each trunk drainage sewer condition indicated that the peak flow reduction was the greatest at 40% and 60% dimensionless upstream area ratio (DUAR) for 1000–5000 m$^3$ and 10,000 m$^3$, respectively. High hydrological applicability under the same rainfall conditions was demonstrated consequent to analyzing the applicability of the installation of a typical reduction facility and trunk drainage sewer.

**Keywords:** dense urban area; flood damage; reduction; trunk drainage sewer system; applicability analysis

## 1. Introduction

According to the Emergency Events Database, floods were the most frequent form of natural disaster in 2019, and typhoons had the highest impact in terms of human casualties and economic damage. In Asia, the number of deaths and people affected by floods and typhoons was the highest among natural disasters, causing severe economic damage accounting for 97.4% of the total. In Japan, a typhoon on 12 October 2019 caused severe economic damage ($17 billion), and in India, floods on 14 July 2019 killed 1900 people. Seven out of nine disasters causing the worst economic damage in Asia in 2019 were floods and typhoons in China [1].

According to a 2020 Organisation for Economic Co-operation and Development report, one in five people living in cities, or 613 million people, are exposed to a 100-year flood. While 70% of the entire city is unexposed, 6% is at risk of being completely submerged. Most of the cities with the largest populations exposed to a 100-year flood are located in Asia, as the density of cities in Asia is the highest considering the distribution of large cities [2].

Urban flooding is considerably different from rural flooding in that an enormous amount of water flows into a receiving reservoir, increasing peak flooding by 1.8–8 fold and the flood volume by up to 6-fold (India National Disaster Management Authority). Urban areas are densely populated, and people living in vulnerable areas suffer from floods, with frequent casualties. Further, secondary effects, such as being exposed to infection, are also damaging in terms of loss of life and livelihood. Furthermore, urban areas are centers of economic activity with important infrastructure. Damage to critical infrastructure in large

cities may have global, not just national, impacts. Therefore, urban flood management must be prioritized [3].

Natural streams and waterways are formed by the force of water flowing from each watershed over thousands of years. Dwellings along rivers and waterways began to grow into villages and cities. This resulted in an increase in water flow in proportion to the urbanization of the watershed. Ideally, natural drains should be widened to accommodate for excessive water flows, such as storms and torrential rain, similar to widening roads for increasing traffic flow; however, residential-priority land use has reduced the capacity of natural drains, which creates the conditions for large-scale flooding. The causes of urban flooding include rainfall concentrated in a specific time period due to the typical climate, increased precipitation due to the urban heat island effect, and poor drainage due to the decrease in the ratio of permeable to impervious areas. Problems associated with urban flooding range from relatively localized events to major events, resulting in cities being flooded for hours to days. The increase in urban flooding is a universal phenomenon and poses great challenges to urban planners around the world [3].

Drainage systems in urban areas are designed based on a specific frequency, but the design frequency is not high. When heavy rainfall occurs, drainage systems in urban areas can very easily become overwhelmed. Drainage systems are also poorly maintained and often fail to meet their designed capacity. Efforts are being made to reduce domestic water inundations, including drainage pipe improvement projects, pumping station efficiency improvements and expansion projects, and underground storage tank installations. In large densely-populated cities, it is difficult to secure a budget to replace old drainage pipes with new ones, and since most of the development has been completed, it is not easy to select a site to install an out-of-region reservoir.

It is difficult to prepare measures to reduce flooding in areas frequently affected by floods due to existing buildings, roads, and various infrastructures. A possible alternative is to install an underground flood-reduction facility, but it is difficult to secure a substantial space due to the high density of underground space. This study aimed to determine whether it is possible to apply a trunk drainage sewer system facility with an optimal flood reduction effect using only a small underground space.

Lee et al. proposed a new operation method linking a centralized reservoir (CR) and a decentralized reservoir (DR) by sharing water level information in the monitoring node of the urban drainage system. CRs yielded worse results than the existing drainage system operation and DR when the water levels of the monitoring node were as high as 1.4–1.5 m and as low as 0.8–0.9 m, respectively, suggesting the need for a cooperative operation plan customized for the situation of the urban drainage system [4].

Qin argued that urban flooding could be mitigated by significantly weakening the peak flow during heavy rains and extending the discharge duration. A new leak control device (that is, leak tanks) was proposed to attenuate the peak flow during a flood and to extend the runoff duration [5].

Kim and Kang proposed water tanks, permeable pavements, and ecological waterways as flood risk reduction facilities that can be installed in dense urban areas, such as Seoul. However, the volume of the water tanks, permeable pavements, and ecological waterways that can be installed in Seoul is 776,588, 89,049, and 81,986 $m^2$, respectively [6].

Alves et al. combined pervious pavements, rainwater barrels, open detention basins, and pipes in six different ways to respond to urban flood risks. This combination was tested to compare green-blue measures with conventional (or gray) measures. A combination of measures can maximize efficiency, with some grey measures focusing on flood risk reduction and some green-blue measures providing public protection [7].

Liu et al. proposed retrofitting green roofs as a promising approach to prevent flooding in densely developed cities. Green roof systems can retain approximately 41–75% of precipitation for two hours, and improved green roof systems reduces the local 10- and 100-year precipitation by 82 and 28%, respectively, as compared to that of a traditional roof system. In continuous simulations, green roof systems also improved evapotranspiration, accounting for 39% of annual precipitation, thereby reducing the cumulative surface runoff [8].

The blue-green drainage infrastructure in Hong Kong can reduce surface runoff, water pollution, heat island effect, carbon footprint, and energy consumption while incorporating the natural water environment into the city. It also complements existing drainage systems by reducing surface runoff and mitigating peak flows, increasing the resilience of the entire drainage system against unexpected extreme events. Considering the scarcity of land resources in Hong Kong, the design of the blue-green drainage infrastructure could possibly be integrated with other public facilities to use the same land, opening up drainage reserves for public enjoyment [9].

This study aims to determine the necessity of a reduction facility combined with an existing drainage system to reduce flooding damage in dense urban areas where the amount of runoff is increasing even under the same precipitation levels and to induce smooth drainage. Therefore, this study proposes the trunk drainage sewer concept that can be operated with the existing drainage system and seeks to analyze its applicability. First, the study conceptually explains the trunk drainage sewer as compared to a typical reservoir and then presents the installation conditions. Then, the input data of the analysis model is calibrated and tested for three dense urban areas where actual flooding has occurred. The applicability of the trunk drainage sewer is analyzed by comparing it with the alternative flood damage reduction measures (reduction facility installation project) actually applied in dense urban areas.

## 2. Materials and Methods

### 2.1. Trunk Drainage Sewer System

The trunk sewer is a sewage pipe used to receive sewage from many tributary branches and sewer lines and to serve as an outlet for a wide area, such as a river, or to distribute it to an intercepting sewer pipe. As such, trunk sewers constitute an important part of the sewage network serving a large population or an industrial center [10].

Regarding the operation of existing drainage systems, when flooding occurs due to rainfall exceeding the design frequency of the drainage pipe and rainwater pumping station, the installation of an underground rainwater storage tank or rainwater runoff reduction facility can reduce flood damage in the downstream area. However, direct discharge of rainwater may be most effective when a river for drainage is located nearby. However, in dense urban areas where direct discharge is difficult due to the absence of adjacent streams for drainage or a river water level higher than the planned flood level, trunk lines are used to allow rainwater to run into an area where the drainage flow is relatively smooth, thereby reducing flood damage.

A trunk drainage sewer system can be defined as a facility that delays runoff using a small-scale underground reservoir and then diverts the rainwater to a drainage pipe in an area with smooth flow using an arterial pipeline.

As shown in Figure 1, the trunk drainage sewer system can be used to improve urban areas susceptible to increasing flood damage from excess runoff that occurs with rainfall exceeding the design frequency. It is expected to reduce the local flood damage risk in situations where typical measures, such as the installation of an underground storage tank and drainage pipe improvement projects, cannot be applied due to restrictions in urban planning and financial conditions, such as site acquisition and lack of project funds.

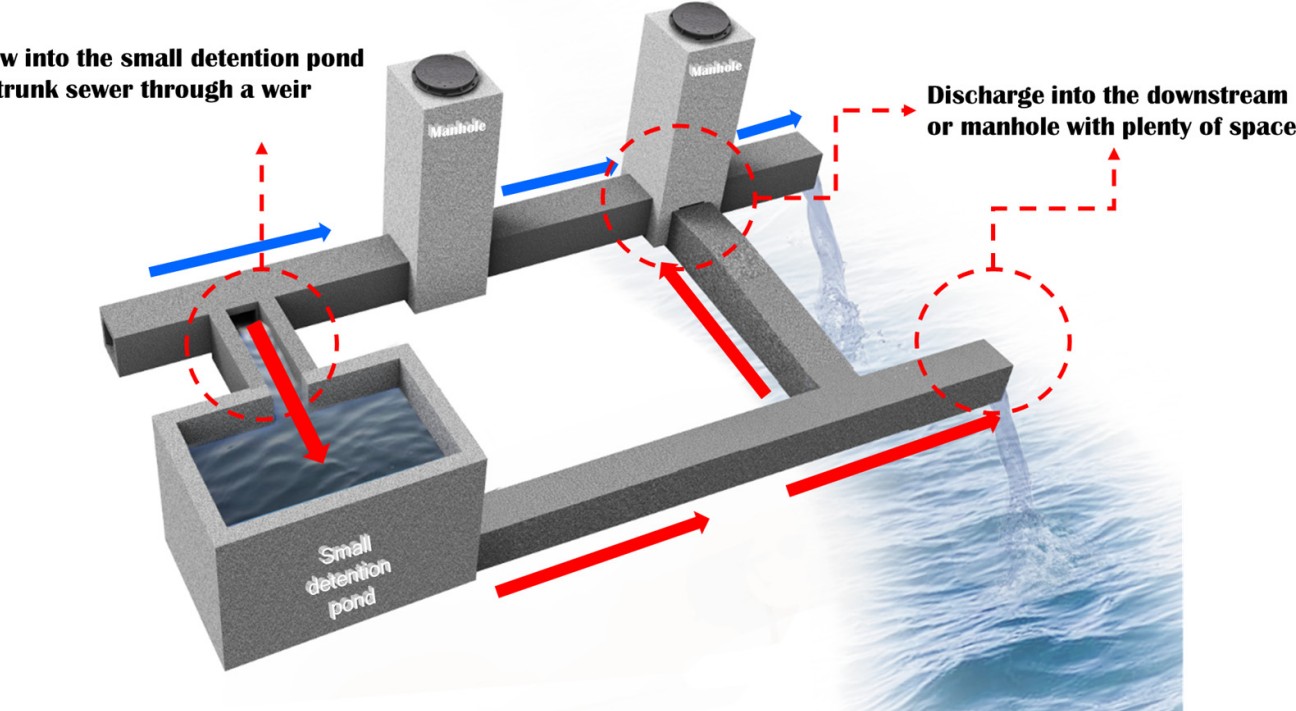

**Figure 1.** Schematic of the trunk drainage sewer system.

The installation conditions and application limits of the arterial reservoir are as follows:
(1) A trunk drainage sewer system, which has a relatively small capacity as compared to
general storage facilities, should be installed in a location where there is no interference
from underground structures. (2) A trunk drainage sewer system should be installed in the
upstream part of the section where delays of waterflow occur or where overflow is frequent,
and after storing a certain amount of flow, it allows rainwater to naturally flow down
to a drainage point (pipe or river) with a smooth flow. (3) The discharge capacity of the
existing drainage pipe is confirmed by applying rainfall conditions of various frequencies
with an urban runoff model such as the stormwater management model (SWMM), and
the capacity of trunk drainage is designed to exceed the discharge capacity of existing
drainage pipes. Under all rainfall conditions, a drainage pipe or stream with a smooth
flow is determined as the discharge point of the trunk drainage sewer system. (4) An
easily-maintained monitoring device should be installed to prevent the inflow of soil and
dirt into the trunk drainage sewer system and to monitor the water level of the trunk
drainage sewer system. (5) A trunk drainage sewer system can be used as an alternative
means to relieve flood damage in the area.

Figure 2 compares and explains the drainage method of typical underground storage
tanks and trunk drainage sewer systems. A typical underground storage tank mainly
drains the rainwater stored in the underground storage tank to the outside using a pump
after rainfall or flooding have ended, as shown in Figure 2a. The trunk drainage sewer
system (Figure 2b) has the advantage of a relatively large storage effect at a smaller capacity
compared to a typical underground storage tank by directly discharging the inflow of
rainwater to a drainage point (pipe or river) in the downstream area where the flow is
natural and smooth, and the discharge capacity is sufficient.

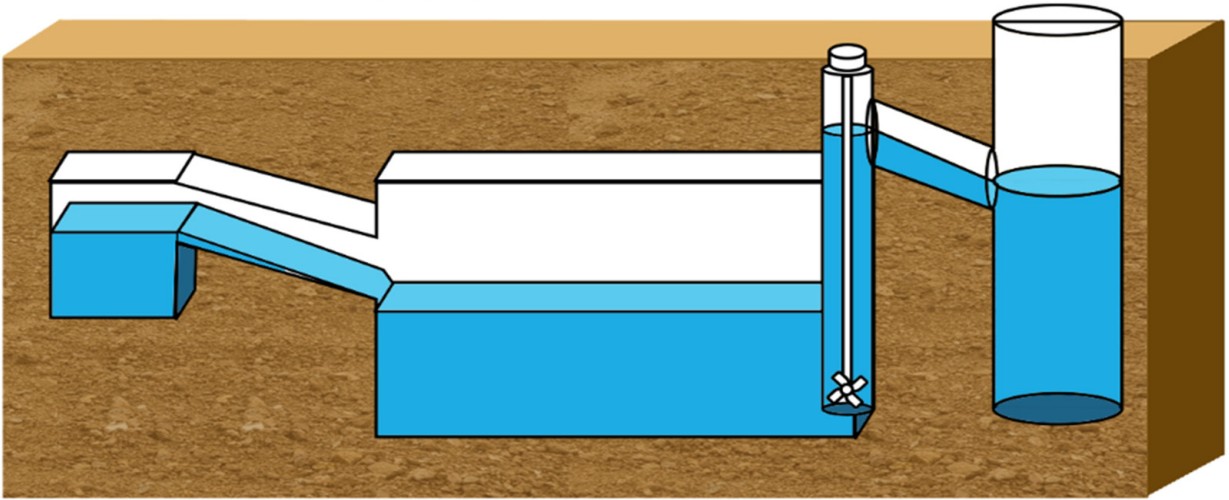

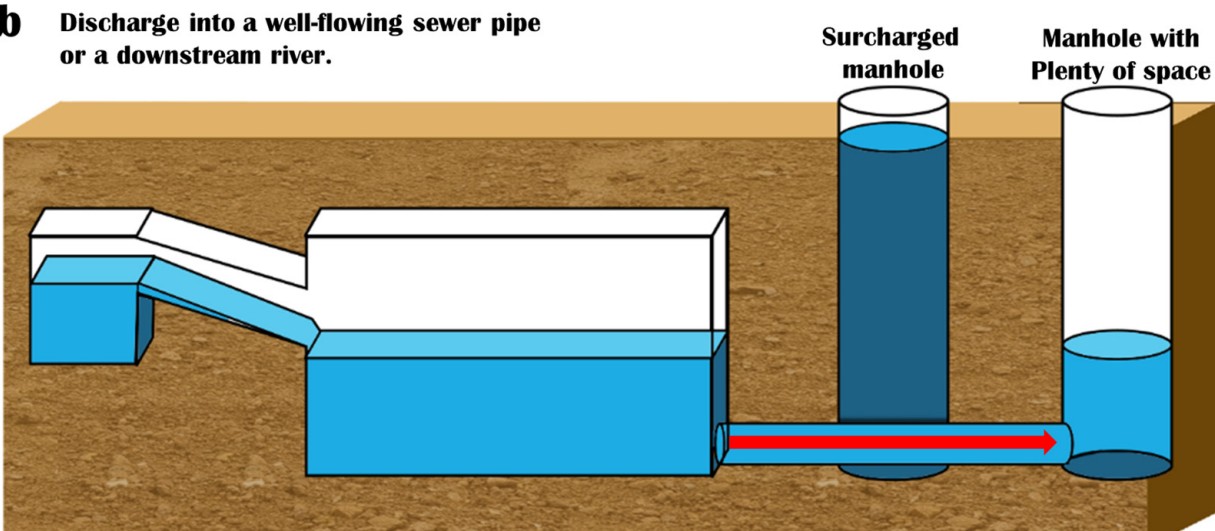

**Figure 2.** Comparison of the discharge plans of the underground storage tank and the trunk drainage sewer system. (**a**) Underground storage tank and (**b**) trunk drainage sewer system drainage methods.

*2.2. Analysis Model (XP-SWMM)*

SWMM, developed by the United States Environmental Protection Agency in 1971, was used to simulate rainfall-runoff processes and to investigate the first and largest flood nodes. SWMM was developed to simulate flow and water quality in urban drainage systems. SWMM can be used for planning, analysis, and design related to storm-water runoff, integrated/sanitary sewers, and other drainage systems in urban and rural areas. It is a comprehensive model that can be used to simulate rainfall events, surface and sewer runoff, track runoff in sewer pipelines, and calculate capacity and volume [11,12]. The SWMM flow routing options include steady flow routing, kinematic wave routing, and dynamic wave routing.

XP-SWMM is a comprehensive one-/two-dimensional modeling software developed by XP Solutions. Hydrological simulations in XP-SWMM can use historical rainfall data or designed storm events, and the hydraulic simulation of XP-SWMM includes flow paths for sewage conduits in dendritic and loop networks. The XP-SWMM can simulate a dynamic stormwater flow path through the drainage system to the drainage point. The simulation model requires two modules, namely, hydrological and hydraulic simulations.

Hydrological simulations require watershed characteristics and precipitation data, while hydraulic simulations require data such as displacement, size, slope, and elevation [13,14].

### 2.3. Sites for Applying the Trunk Drainage Sewer System

The watersheds with a history of flooding to which the trunk drainage sewer system was applied in this study are the Gunja watershed, located in Jungnang-gu, Seoul; the Dorim watershed, located in Yeongdeungpo-gu, Seoul; and the Dowon watershed, located in Yeosu-si, Jeollanam-do (Figure 3). The Dowon watershed is characterized by two main drainage pipes merging into one before being discharged.

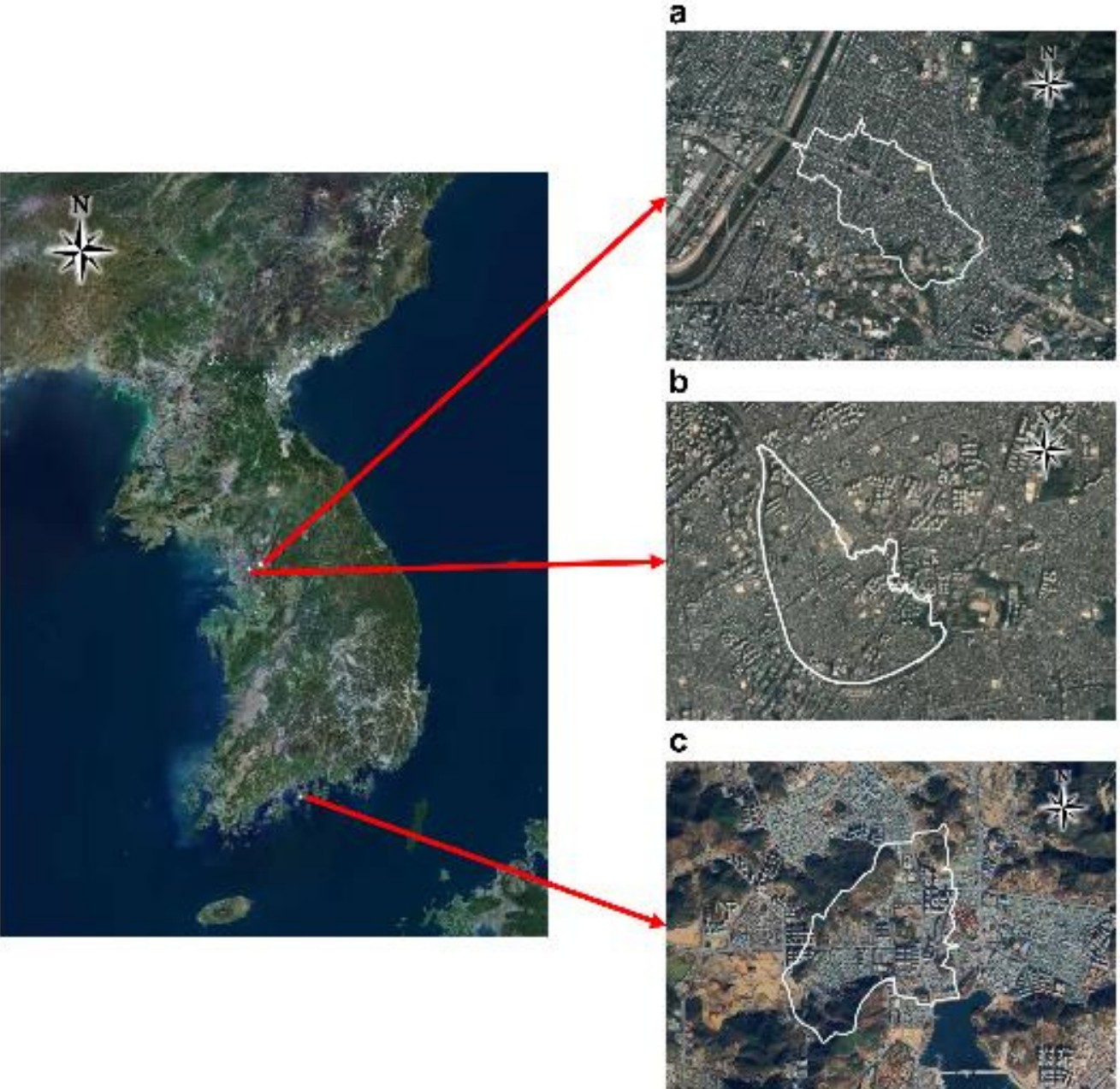

**Figure 3.** The watersheds selected for this study. (**a**) Gunja, (**b**) Dorim, and (**c**) Dowon watersheds in Seoul, South Korea.

### 2.3.1. Gunja Watershed

The Gunja watershed is located on the lower left bank of the Jungnangcheon stream. The watershed area is 0.966 km$^2$ with a river length of 2.15 km, and approximately 75% of the total watershed area is impervious. The Gunja area has a history of flood damage due to heavy rains that occurred from 20 to 21 September 2010, and the measured inundation area was 0.192 km$^2$, accounting for 19.8% of the total area (Figure 4).

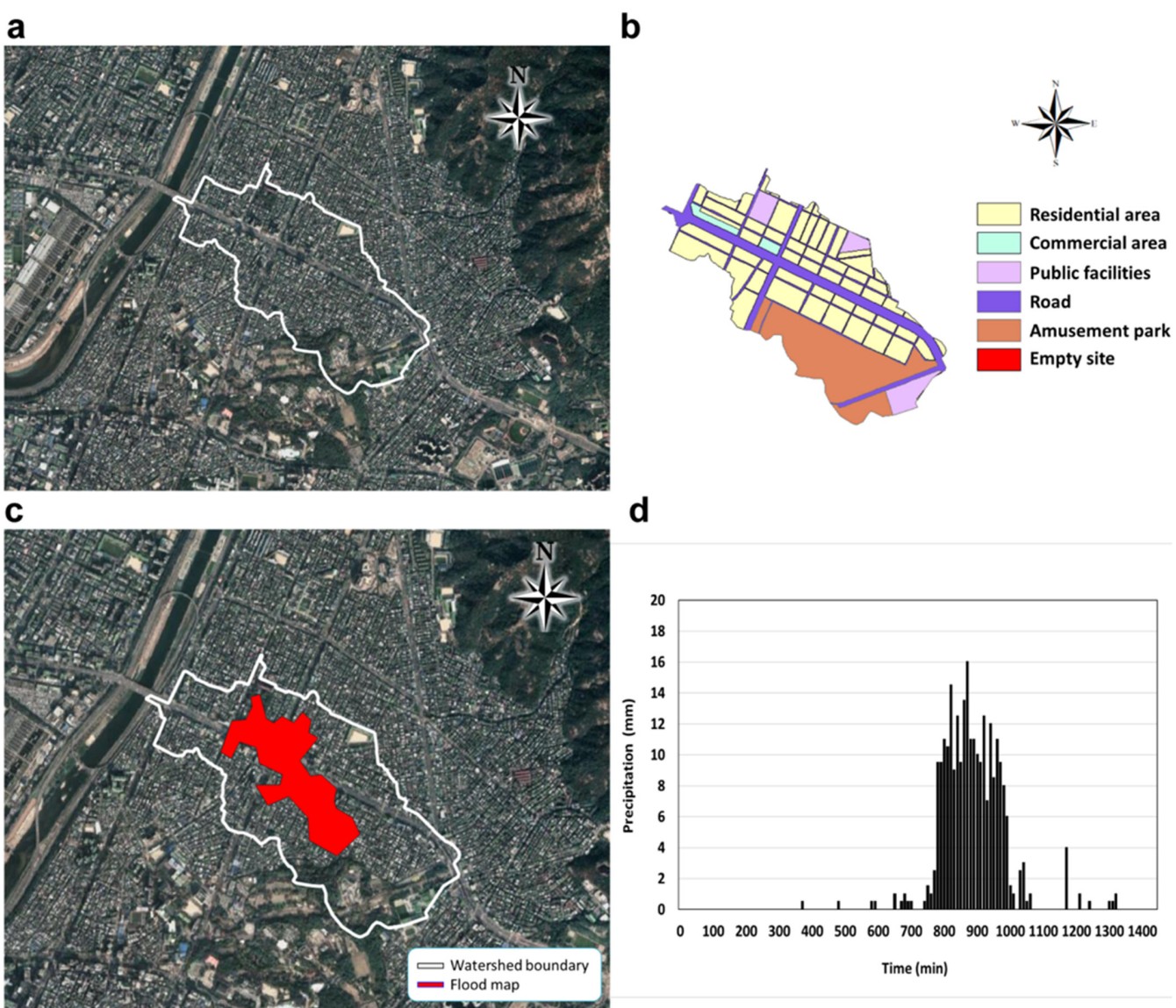

**Figure 4.** (**a**) Watershed boundary, (**b**) land use status, (**c**) flood damage history, and (**d**) damage-causing rainfall (20 to 21 September 2010) in the Gunja watershed, Seoul.

### 2.3.2. Dorim Watershed

The Dorim watershed was selected as a special management area among the areas susceptible to flooding as designated by the Seoul Metropolitan City. It is still considered a key management area. The watershed area is 2.71 km$^2$ with a river length of 1.93 km, and the impervious area in terms of land use accounts for 88% of the total watershed area, which is higher than that of the Gunja watershed. The Dorim watershed, which is vulnerable to flooding, has two drainage pumping stations, namely, Daerim 2 and Daerim 3, to reduce flood damage. The torrential rain that occurred 21–22 September 2010 caused 0.356 km$^2$ of flood damage (Figure 5).

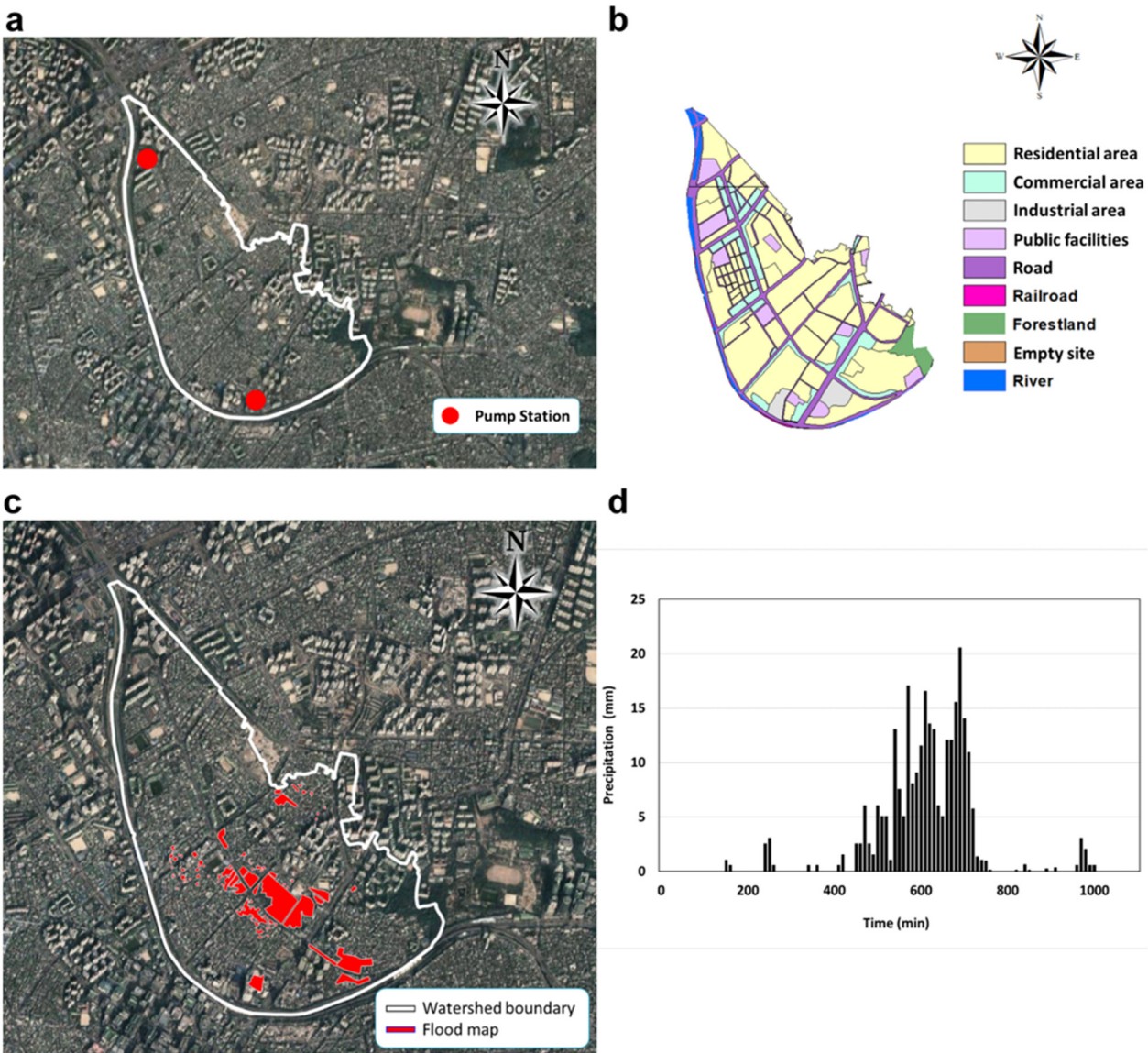

**Figure 5.** (**a**) Watershed boundary, (**b**) land use status, (**c**) flood damage history, and (**d**) damage-causing rainfall (21 to 22 September 2010) in the Dorim watershed, Seoul.

### 2.3.3. Dowon Watershed

The Dowon watershed of Yeosu includes Seonwon-dong, Ansan-dong, and Hak-dong in Yeosu-si, Jeollanam-do. The watershed area is 23.3 km$^2$, with a river length of 1.91 km. It is surrounded by mountains, and the rainwater discharges into the sea, which is located to the south. There is a history of habitual flood damage, namely, torrential rain in 2010, 2016, and 2017, and typhoon "Muifa" in 2011. In particular, the torrential rain that occurred from 19 to 23 August 2017 caused flood damage over 0.214 km$^2$ from the sea discharge location to the residential areas surrounding the main crossroads in the Dowon watershed (Figure 6).

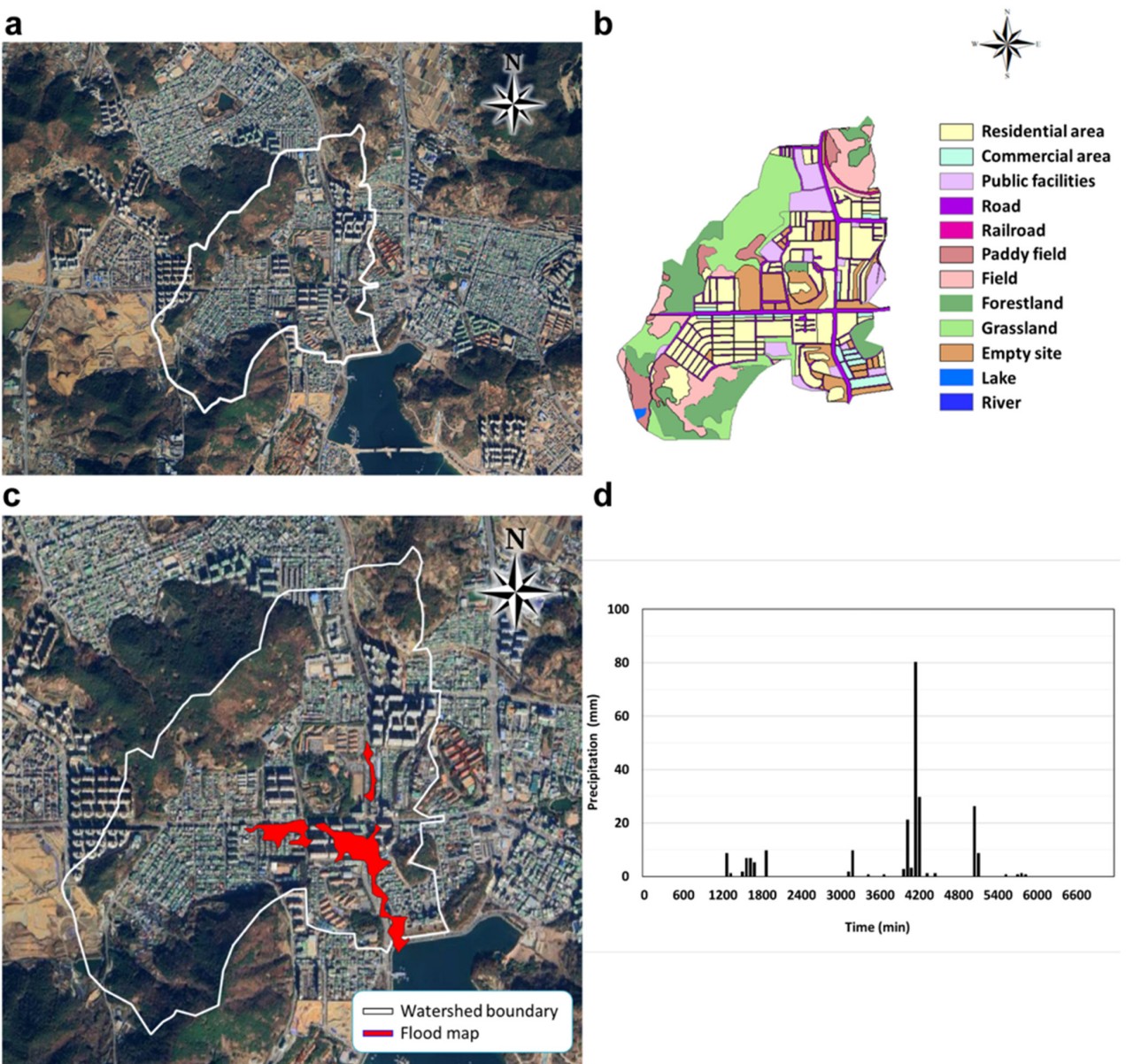

**Figure 6.** (**a**) Watershed boundary, (**b**) land use status, (**c**) flood damage history, and (**d**) damage-causing rainfall (19 to 23 August 2017) in the Dowon watershed, Yeosu.

### 2.4. LSSI Technique

In order to judge the accuracy of the inundation simulation, it is important to measure the spatial location accuracy of the inundation range and to compare the inundation area. The LSSI technique is a method of measuring spatial location accuracy by calculating the overlapping area of the inundation trace and the simulated flood range. In this study, the accuracy of the inundation simulation results was measured using the LSSI technique. The LSSI technique uses values calculated in exponential form to classify the positional accuracy of the reference and comparison data into values between 0 and 1, with values closer to 1 indicating higher spatial position accuracy [15]. The LSSI technique classifies the evaluation criteria from "Fail" to "Excellent", as shown in Table 1, based on the range of values calculated using Equation (1).

$$\text{LSSI } (\%) = \frac{A \cap B}{A \cup B} \times 100,$$

(1)

where A is the inundation trace map and B is the simulated flood analysis results.

**Table 1.** Evaluation criteria for the LSSI technique.

| LSSI Range | Degrees of Accuracy |
|:---:|:---:|
| 5.0< | Fail |
| 10.0< | Poor |
| 20.0< | Fair |
| 30.0< | Good |
| 40.0< | Excellent |

## 3. Results and Discussions

### 3.1. SWMM Model Application

The input data for the SWMM model was constructed using regional sewer pipe data and a numerical topographic map. The runoff data for the Gunja watershed measured by the Seoul Waterworks Research Institute was used to calibrate the input data of the SWMM model. The pump inflow data from the Daerim 3 drainage pump station was used as the SWMM model input data for the Dorim watershed. As there was no actual flow data for the Dowon watershed, the method of estimating input parameters based on the flood damage area was adopted (Table 2 and Figure 7). The data measured at the observation point of the Automated Synoptic Observing System of the Korea Meteorological Administration (Seoul Meteorological Observatory for the Gunja and Dorim watersheds and Yeosu Meteorological Observatory for the Dowon watershed) at the time of the flood damage was used for the rainfall event for model input data calibration, estimation, and testing.

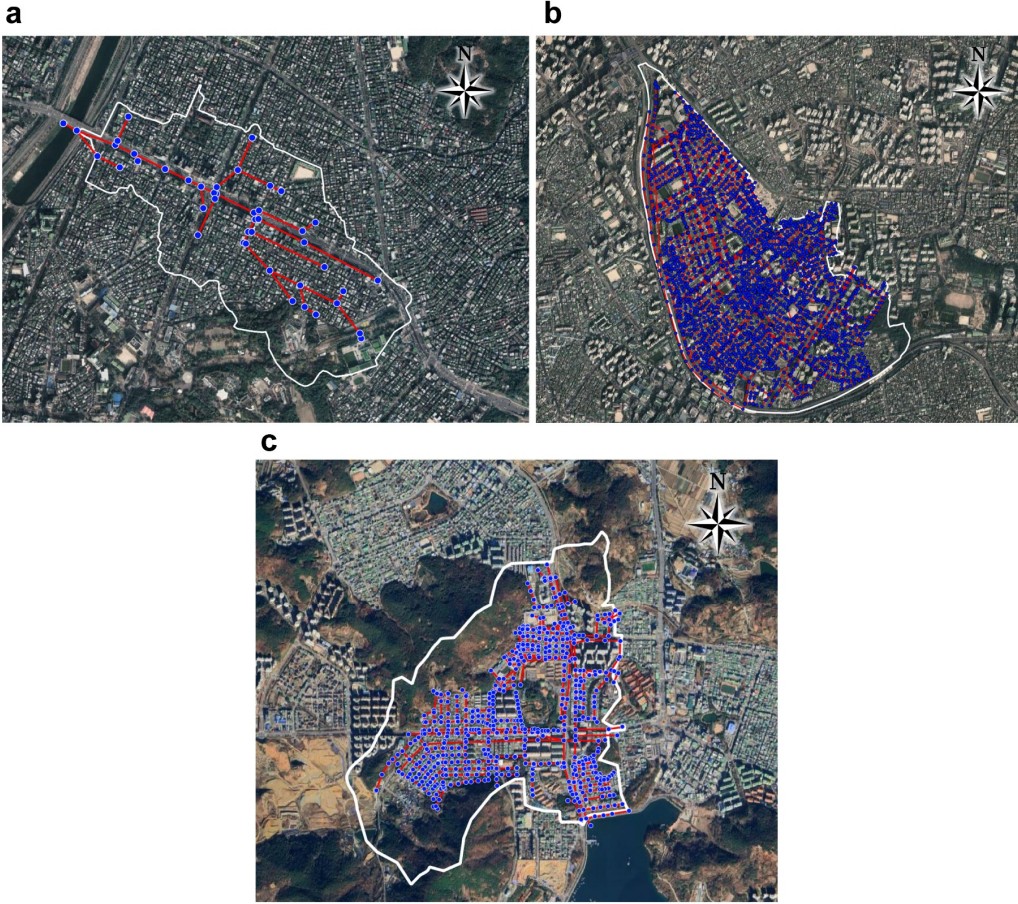

**Figure 7.** SWMM model input data configuration for the (**a**) Gunja, (**b**) Dorim, and (**c**) Dowon watersheds.

**Table 2.** SWMM model input data configuration.

| Input Parameters | Gunja Watershed | Dorim Watershed | Dowon Watershed * |
|---|---|---|---|
| Roughness coefficient of the impervious area | 0.01–0.040 | 0.014–0.015 | 0.013–0.030 |
| Surface storage in the impervious area (mm) | 0.0–5.0 | 0.0–5.0 | 0.0–5.0 |
| Initial infiltration (Horton's) (mm/h) | 5.0–15.0 | 2.5–25.4 | 7.0–35.0 |
| Attenuation of infiltration (Horton's) (1/s) | 0.00056 | 2.0 | 0.05 |
| Subcatchment width (m) | 25.0–1207.24 | 2.24–114.02 | 2.51–135.78 |
| Imperviousness (%) | 22.4–100 | 1.6–100 | 5.6–100.0 |
| Subcatchment slope | 0.002–0.108 | 0.000–0.369 | 0.001–0.317 |

Note: * Parameters estimated for the Dowon watershed.

Figures 8 and 9 and Table 3 show the results of applying the calibrated SWMM model input data and those of testing the amount of runoff at the measured points. The runoff hydrograph observed at the outlet point for the rainfall that occurred on June 12, 2010 over the Gunja watershed was compared with the simulated runoff hydrograph. The observed and simulated peak flows at the outlet were 2.077 and 2.321 m$^3$/s, respectively, with an error of 0.244 m$^3$/s, showing an accuracy of 88.25%. The pump inflow volume of the Daerim 3 drainage pumping station for the rainfall that occurred on August 21, 2010 over the Dorim watershed was compared with the simulated runoff hydrograph. The observed and simulated peak flows at the drainage pumping station were 28.52 and 31.22 m$^3$/s, respectively, with an error of 2.7 m$^3$/s, showing an accuracy of 93.9% (Figure 8).

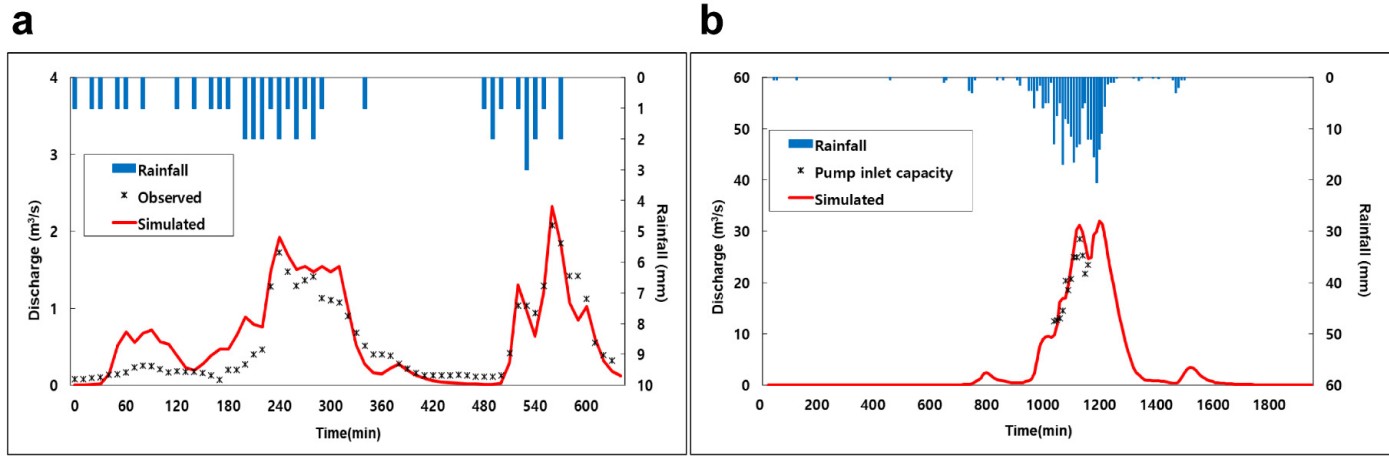

**Figure 8.** Calibration of the simulated results against measured runoff for the (**a**) Gunja and (**b**) Dorim watersheds.

Figure 9 shows that the simulated flood-damaged area in the Gunja watershed was 0.1852 km$^2$, which showed only an error of 0.0068 km$^2$ when compared with the observed inundation area of 0.192 km$^2$. In contrast, the simulated inundation area in the Dorim and Dowon watersheds was 0.005 and 0.007 km$^2$, respectively, which was larger than the actual inundation area. The LSSI, which compares the spatial distribution of the inundation area, had an accuracy of 53.25% in the Gunja watershed, where inundation was concentrated; in the Dorim watershed it had a 32.27% accuracy, where inundation was scattered, with their qualitative evaluation of "Good" and "Excellent," respectively (Table 3).

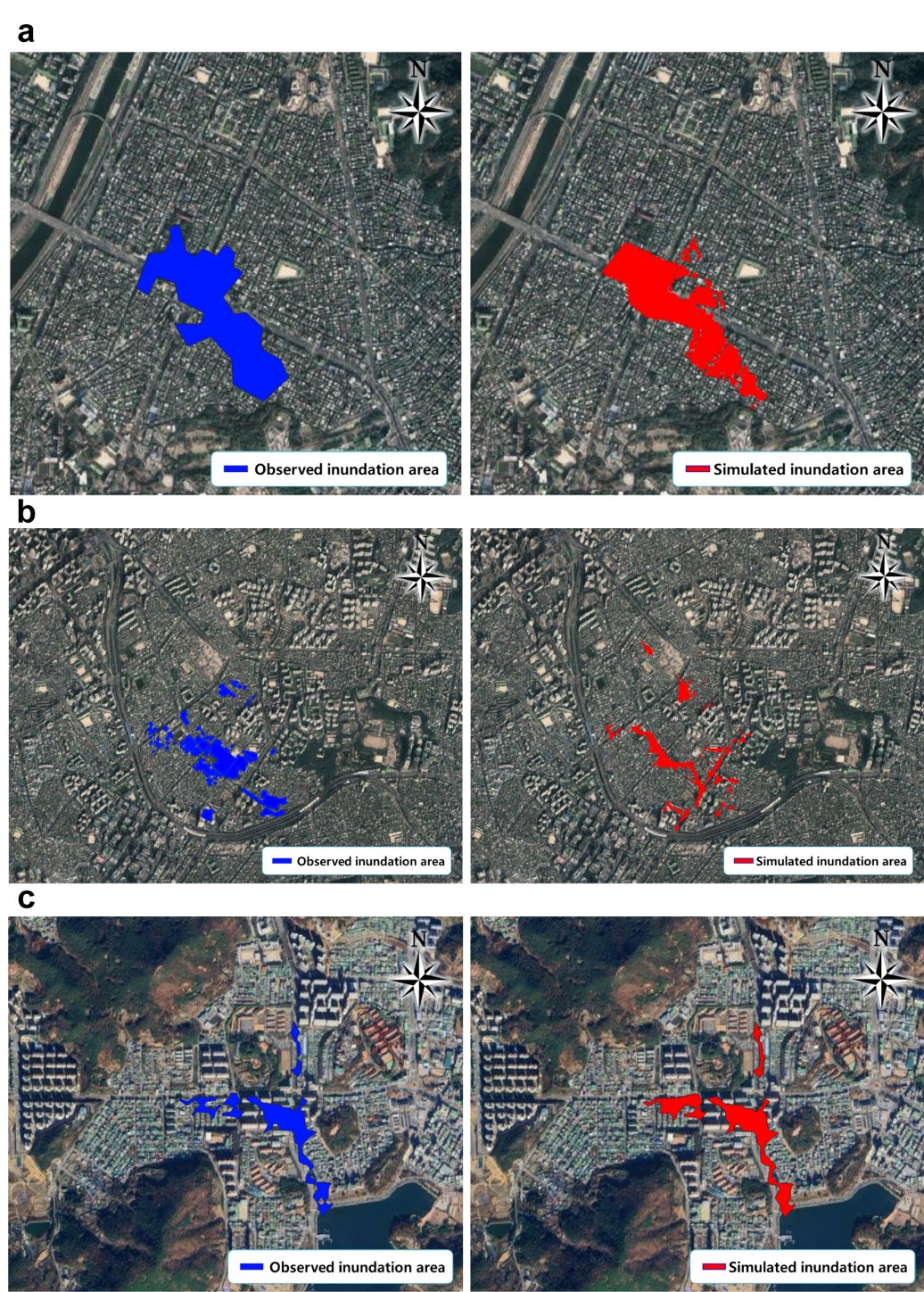

**Figure 9.** Comparison of inundation trace and simulated flood range in the (**a**) Gunja, (**b**) Dorim, and (**c**) Dowon watersheds.

**Table 3.** Comparison of the actual inundation area with the simulated result.

| Watershed | Actual Inundation Area (km²) (A) | Simulated Inundation Area (km²) (B) | A ∪ B (km²) | A ∩ B (km²) | LSSI (%) | Degree |
|---|---|---|---|---|---|---|
| Gunja watershed | 0.192 | 0.185 | 0.240 | 0.128 | 53.33 | Excellent |
| Dorim watershed | 0.356 | 0.361 | 0.587 | 0.195 | 32.22 | Good |
| Dowon watershed | 0.214 | 0.221 | 0.289 | 0.137 | 47.41 | Excellent |

### 3.2. Applicability Review of the Trunk Drainage Sewer System

In order to apply the trunk drainage sewer system as an alternative to the installation of a typical storage facility, the applicability review was conducted using six analysis conditions (1000, 3000, 5000, 10,000, 20,000, and 30,000 m$^3$) that limit the capacity to a maximum of 30,000 m$^3$, as mentioned in the installation conditions. To generally achieve the greatest reduction effect, it is better to install the trunk drainage sewer system in the downstream area where the outflow amount is at its maximum. Considering the merits of the trunk drainage sewer system that can specify the drainage point after storage, the conditions for the installation location were also diversified, and the applicability was further reviewed. As for the installation site of the trunk drainage sewer system, the area ratio of the upstream area of the watershed (hereinafter referred to as the DUAR) was calculated based on the trunk line, which is divided into 20, 40, 60, and 80%. Since two trunk pipelines are joined in the Dowon watershed, the trunk drainage sewer system was installed in a one trunk pipeline, with the DUAR set to only 20 and 40% (Figure 10).

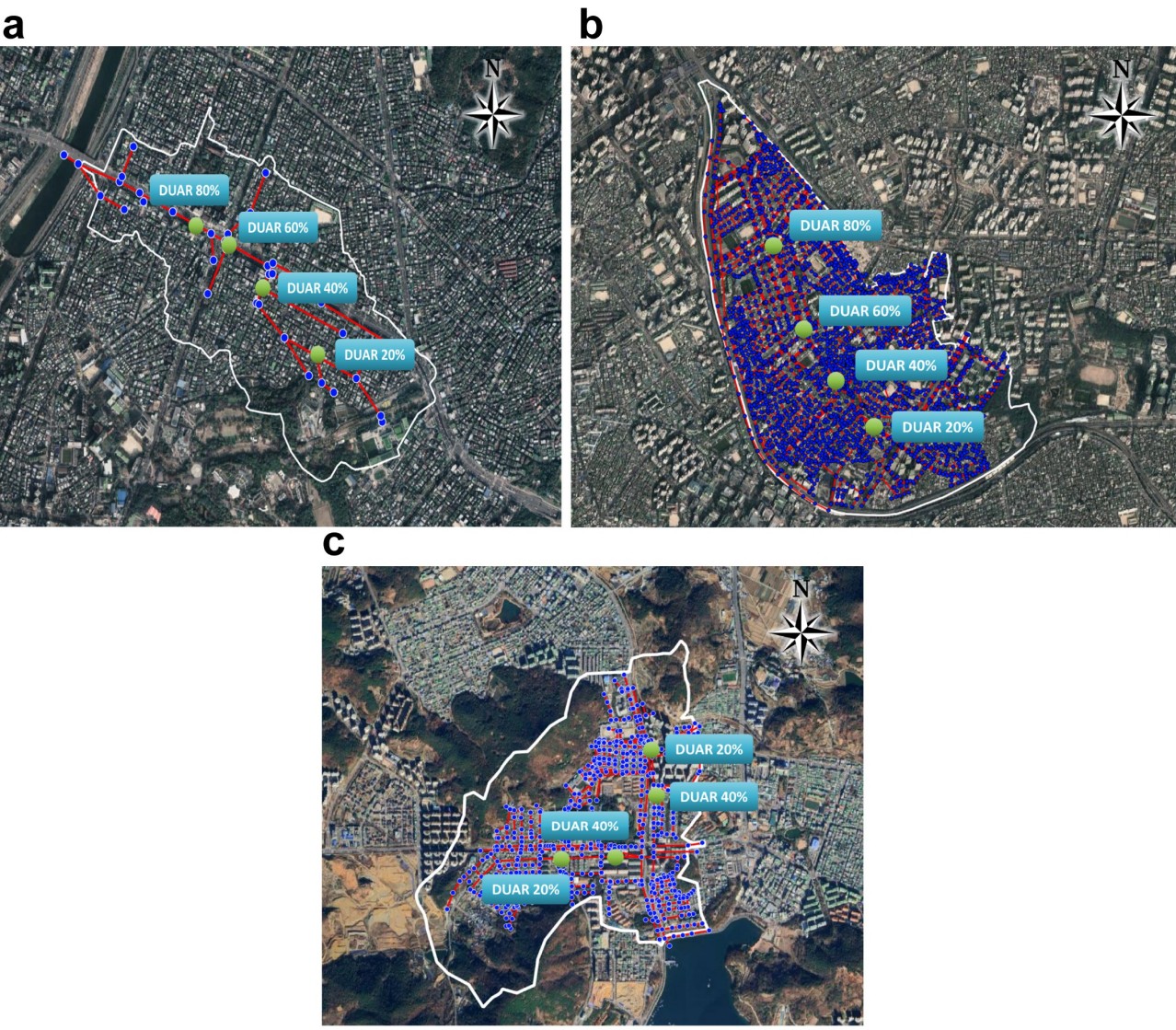

**Figure 10.** Location of the trunk drainage sewer system in the (**a**) Gunja, (**b**) Dorim, and (**c**) Dowon watersheds.

The largest peak flow reduction in the Gunja watershed was 33.29% at the DUAR 40% point when the trunk drainage sewer capacity was 1000–5000 m$^3$. When the trunk drainage sewer capacity was 10,000 m$^3$, the peak flow reduction was the largest (51.03%) at the DUAR 60% point, and at a capacity of 10,000–30,000 m$^3$, the peak flow reduction was the largest (79.38%) at the lowest point, that is, DUAR 80%. The applicability in the Dorim watershed was the same in terms of the capacity of the trunk drainage sewer and the location of the installation site. However, in the Dorim watershed, the capacity of the trunk drainage sewer was relatively small, considering that the runoff volume due to rainfall was 512,700 m$^3$, with a drainage structure letting the runoff flow into the downstream drainage pumping station through numerous branch lines, leading to a peak flow reduction of 0.28–16.06%. In the Dowon watershed, the largest peak flow reduction was 11.08% when installed at the DUAR 40% point. However, as in the Dorim watershed, the capacity of the trunk drainage sewer was relatively small, considering the runoff volume, leading to a peak flow reduction of 0.90–11.08% (Table 4).

**Table 4.** Peak flow reduction by capacity and installation location of the trunk drainage sewer.

| Watershed | Capacity (m$^3$) | Installation Location (DUAR) | | | |
| --- | --- | --- | --- | --- | --- |
| | | 20% | 40% | 60% | 80% |
| Gunja watershed | 1000 | 8.51 | 11.48 | 7.14 | 5.72 |
| | 3000 | 15.78 | 24.99 | 22.48 | 18.86 |
| | 5000 | 18.51 | 33.29 | 33.11 | 29.57 |
| | 10,000 | 19.79 | 44.65 | 51.03 | 48.03 |
| | 20,000 | 21.09 | 47.51 | 71.43 | 72.98 |
| | 30,000 | 21.74 | 47.62 | 74.03 | 79.38 |
| Dorim watershed | 1000 | 0.45 | 0.50 | 0.39 | 0.28 |
| | 3000 | 0.52 | 0.58 | 0.56 | 0.52 |
| | 5000 | 0.57 | 0.66 | 0.65 | 0.61 |
| | 10,000 | 0.65 | 1.41 | 4.76 | 4.05 |
| | 20,000 | 1.22 | 4.36 | 9.16 | 10.36 |
| | 30,000 | 1.53 | 7.57 | 12.53 | 16.06 |
| Dowon watershed | 1000 | 0.90 | 0.96 | – | – |
| | 3000 | 0.94 | 1.07 | – | – |
| | 5000 | 1.00 | 1.22 | – | – |
| | 10,000 | 1.11 | 2.02 | – | – |
| | 20,000 | 1.65 | 5.71 | – | – |
| | 30,000 | 2.06 | 11.08 | – | – |

*3.3. Reduction Effect Using Reduction Measures*

To prepare for repeated flood damage in the watersheds, typical measures, such as the installation of an underground storage tank and the improvement of pipelines, have been planned and implemented. As the reduction effect for the trunk drainage sewer system varies depending on the capacity, location, and setting of the discharge point, it can be applied while changing the analysis conditions in various ways. Because it has the advantage of being easy to actively respond to changes in the installation conditions, such as land expropriation and project cost, in the actual area, this study intended to compare and review whether the arterial reservoir had applicability as an alternative to the installation of a typical abatement facility. As stated in the installation conditions, it is necessary to consider the characteristics of the watershed as the installation capacity, location, and discharge point need to be determined for the installation of the trunk drainage sewer system.

The Seoul Metropolitan Government selected 34 flood-prone areas based on the flood damage caused in 2010 and 2011 and invested approximately KRW 1530 billion in the total project cost from March 2011 to December 2023 to increase the capacity of the drainage facilities, such as sewage and pumping stations. However, due to climate change, extreme weather events, such as flash floods and heavy rainfall that exceed the designed capacity of

drainage facilities, have increased in Seoul. It has become difficult to reduce the likelihood of flood events and associated damage with the Seoul flood control policy and management, which mainly increases the capacity of the drainage facilities. It is necessary to increase the effectiveness of the Seoul flood control policy and management and improve the adaptability to climate change through flood risk assessments based on the characteristics of the watersheds that exacerbate flood damage rather than on the highly uncertain rainfall characteristics [16].

### 3.3.1. Gunja Watershed

The Seoul Metropolitan City, which manages flood damage in the Gunja watershed, established a plan to install a 25,000 m$^3$ underground storage tank in most downstream areas of the watershed [16]. The flood reduction effect was compared by setting the installation conditions for reviewing the applicability of the trunk drainage sewer system in the area at a 5000 m$^3$ capacity and 40% DUAR, as well as setting the discharge point of the trunk line at a 60% DUAR (Figure 11). Additionally, the rainfall conditions of the rainfall event from August 19 to 23, 2017, which caused flood damage in the watershed, were used.

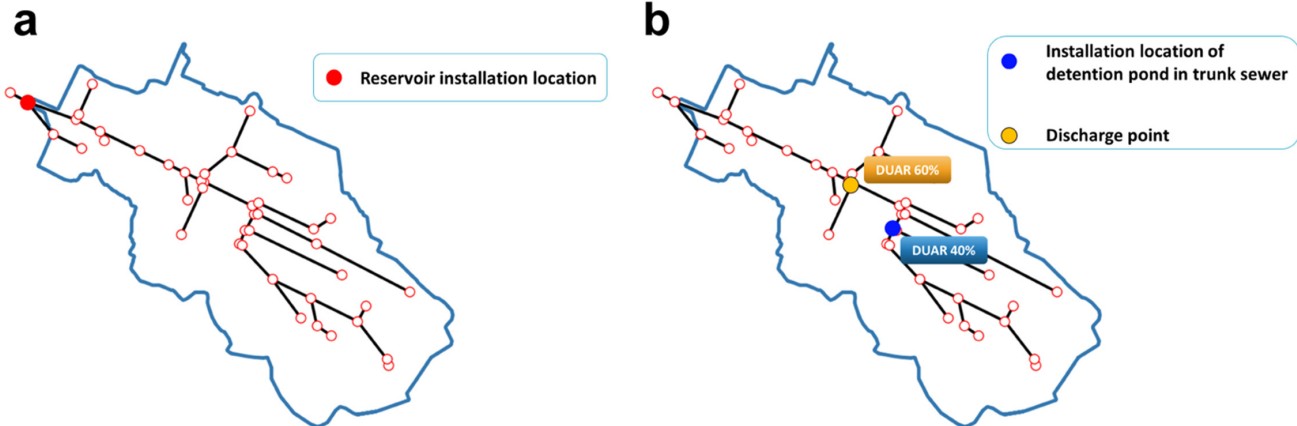

**Figure 11.** Installation conditions for the reduction facilities in the Gunja watershed. (**a**) Installation location of the underground storage tank and (**b**) installation conditions for the trunk drainage sewer.

As shown in Table 5, as a result of installing a 25,000 m$^3$ underground storage tank, which is a typical flood damage reduction measure in Seoul, the number of inundated cells was reduced by 877, the inundation area by 98.956 m$^2$, and the inundation volume by 25,227 m$^3$, as compared to before installation. The spatial distribution of the inundation area in Figure 12 indicates that the inundation of the trunk line was reduced, but the inundation occurring in the branch rainwater conduit located in the northern part of the area was not reduced.

**Table 5.** Flood analysis according to the reduction facility installation conditions (Gunja watershed).

| Structural Improvement Plan | Number of Inundated Cells | Inundation Time (min) | Inundation Area (m$^2$) | Inundation Volume (m$^3$) | Flood Reduction (%) |
|---|---|---|---|---|---|
| Not installed | 1665 | 450 | 185.213 | 50,149 | - |
| Installation of the underground storage tank | 788 | 270 | 86.257 | 24,922 | 53.43 |
| Installation of the trunk drainage sewer | 730 | 225 | 81.253 | 22,796 | 56.13 |

By installing the trunk drainage sewer, the number of inundated cells was reduced by 935, the inundation area by 103,960 m$^2$, and the inundation volume by 27,353 m$^3$, as compared to before installation. Even when compared with the installation of the

underground storage tank, the number of inundated cells was reduced by 58, the inundation area by 5004 m$^2$, and the inundation volume by 2126 m$^3$. The spatial distribution of the inundation area revealed that the inundation area of the rainwater conduit in the northern branch of the area, which had not been reduced by the installation of the underground storage tank, was reduced.

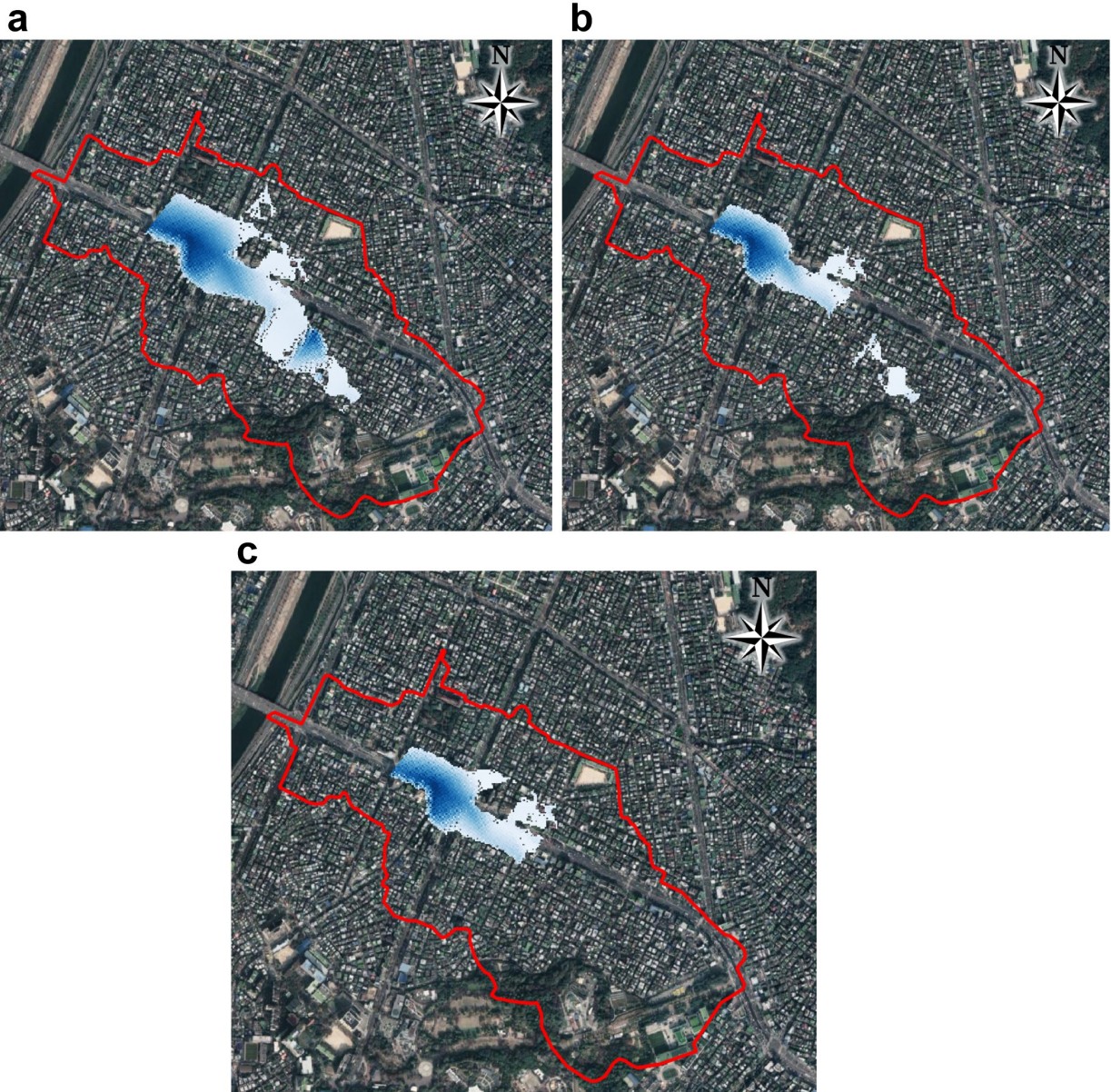

**Figure 12.** Flood analysis according to the reduction facility installation conditions in the Gunja watershed. (**a**) Not installed, (**b**) installation of the underground storage tank, and (**c**) installation of the trunk drainage sewer.

With 1440 min of rainfall duration, the time of inundation was 450 min when the facility was not installed, 270 min when the underground storage tank was installed, and 225 min when the trunk drainage sewer was installed. Due to the storage effect caused by the installation of the underground storage tank, the flooding time of the basin was reduced by 180 min. However, after the detention of the rainfall in the upper stream of the basin, the inundation continued for 45 min longer than the trunk drainage sewer designating the discharge point beyond the congested sewer section.

### 3.3.2. Dorim Watershed

The Seoul Metropolitan City has installed a 30,000 m$^3$ capacity underground storage tank for reducing the flood damage in the Dorim watershed, as was done in the Gunja watershed [16]. Figure 13 shows the installation location of the underground storage tank. According to the peak flow reduction rate analysis conditions in Table 3, the trunk drainage sewer in this study was installed with a capacity of 5000 m$^2$ at DUAR 40%, and the discharge point was set at DUAR 60%. Additionally, the rainfall conditions of the rainfall event from 21 to 22 September 2010, which caused flood damage in the watershed, were used.

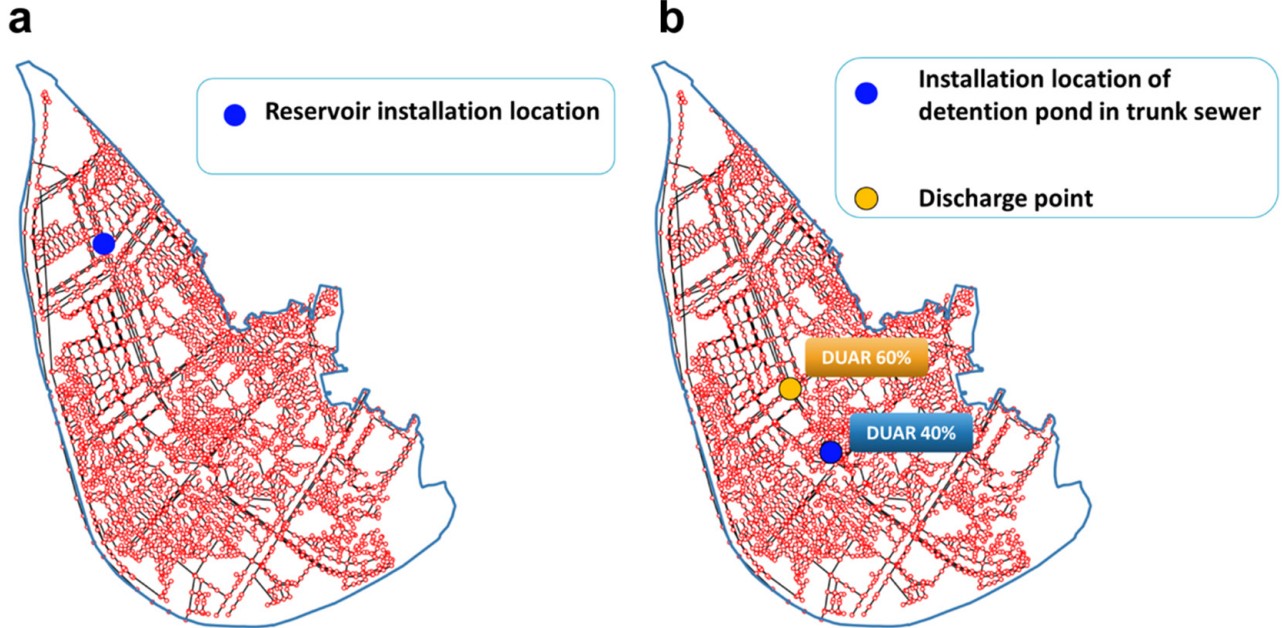

**Figure 13.** Installation conditions for reduction facilities in the Dorim watershed. (**a**) Underground storage tank and (**b**) trunk drainage sewer.

As shown in Table 6, as a result of installing a planned 30,000 m$^3$ underground storage tank by Seoul Metropolitan City, the number of inundated cells was reduced by 21, the inundation area by 2594 m$^2$, and the inundation volume by 11,098 m$^3$, compared to before installation. The spatial distribution of the inundation area in Figure 14 showed no significant difference from before the installation of the reduction facility.

**Table 6.** Flood analysis in the Dorim watershed according to the reduction facility installation conditions.

| Structural Improvement Plan | Number of Inundated Cells | Inundation Time (min) | Inundation Area (m$^2$) | Inundation Volume (m$^3$) | Flood Reduction (%) |
|---|---|---|---|---|---|
| Not installed | 357 | 370 | 44,108 | 303,209 | - |
| Installation of the underground storage tank | 336 | 400 | 41,514 | 292,111 | 5.88 |
| Installation of the trunk drainage sewer | 318 | 320 | 39,290 | 265,650 | 10.92 |

By installing the trunk drainage sewer, the number of inundated cells was reduced by 39, the inundation area by 4818 m$^2$, and the inundation volume by 37,559 m$^3$, as compared to before installation. When compared to the installation of the underground storage tank, the number of inundated cells was further reduced by 18, the inundation area by 2224 m$^2$, and the inundation volume by 26,461 m$^3$. The spatial distribution of the inundation area indicates a reduction in the flood range centered on the installation site of the trunk drainage sewer. However, under all conditions, that is, with no reduction facility, no trunk drainage

sewer installed, and an underground storage tank installed, the flood damage caused by the insufficient capacity of the pumping station in the Dorim watershed was not resolved, requiring a more fundamental solution, such as the rehabilitation of the sewer system.

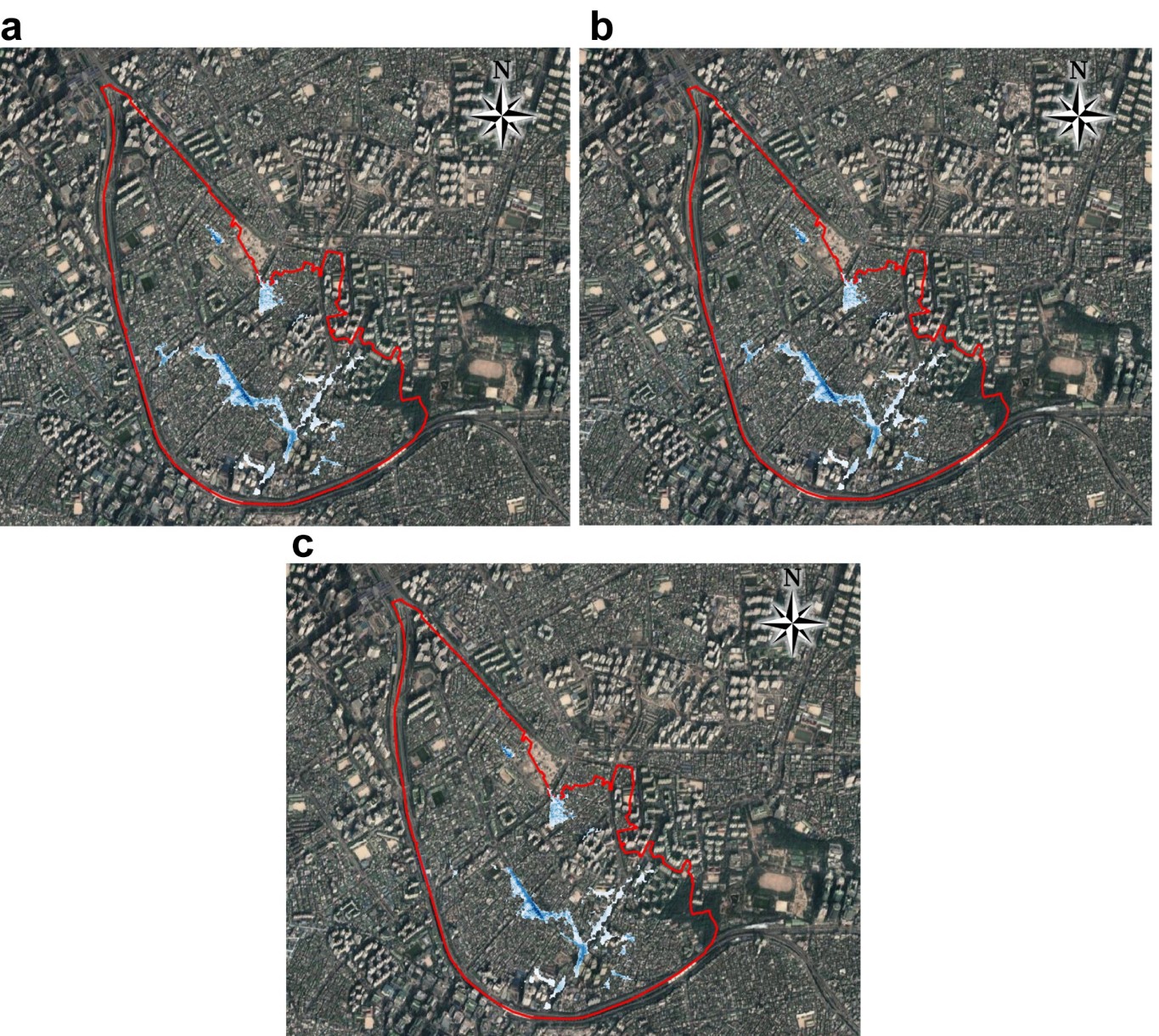

**Figure 14.** Flood analysis in the Dorim watershed according to the reduction facility installation conditions. (**a**) Not installed, (**b**) installation of the underground storage tank, and (**c**) installation of the trunk drainage sewer.

With 980 min of rainfall duration, the time of inundation was 370 min when the facility was not installed, 400 min when the underground storage tank was installed, and 320 min when the trunk drainage sewer was installed. After installing an underground storage tank, the inundation time increased by 30 min. Furthermore, when trunk drainage sewer designating a discharge point was installed outside the congested sewer section after detention in the upper stream of the basin, inundation time was reduced by 50 min.

### 3.3.3. Dowon Watershed

In the case of the Dowon watershed, a plan to improve the rainwater conduit with insufficient water flow in the Dowon intersection area proposed in the "Establishment of Sewerage Maintenance Measures in the Dowon Intersection" has been promoted (Table 7) [17].

**Table 7.** Dowon watershed rainwater conduit improvement plan.

| Division | Length (m) | Rainwater Conduit | |
|---|---|---|---|
| | | Existing | Improved |
| Total | 1651.65 | – | – |
| ① | 244.11 | Box 1.5 m × 1.5 m | Box 2.0 m × 2.0 m |
| ② | 285.53 | D1000 mm | D1200 mm |
| ③ | 421.50 | D600 mm | D900 mm |
| ④ | 258.52 | D500 mm | D1200 mm |
| | 173.25 | D800 mm | |
| ⑤ | 268.74 | D700 mm | D1200 mm |

A 3000 m$^3$ trunk drainage sewer was installed at the DUAR 40% point in the Dowon watershed, and the junction of the two trunk lines was set as the discharge point (Figure 15). Additionally, the rainfall conditions of the rainfall event from August 19 to 23, 2017, which caused flood damage in the watershed, were used.

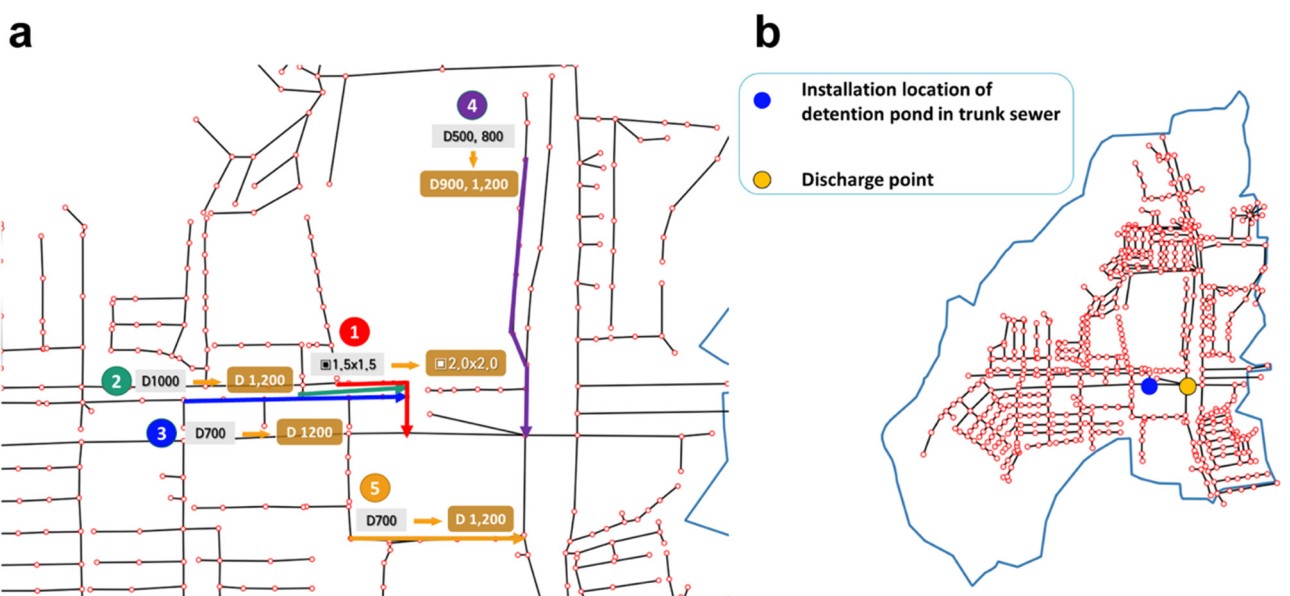

**Figure 15.** Installation conditions for the reduction facilities in the Dowon watershed. (**a**) Conduit improvement plan and (**b**) trunk drainage sewer.

As shown in Table 8, no inundation occurred after improving the rainwater conduit in the Dowon watershed, according to the "Establishment of Sewerage Maintenance Measures in the Dowon Intersection" [17]. By installing the trunk drainage sewer, the number of inundated cells was reduced by 1451, the inundation time by 210 min, the inundation area by 144,996 m$^2$, and the inundation volume by 13,472 m$^3$, as compared to before installation. In view of the spatial distribution of the non-installation and inundation locations, the decrease in the inundation at the discharge point to the sea may be due to smooth drainage with a decrease in the discharge volume at the discharge point of the trunk drainage sewer (Figure 16). Further analysis is required according to the oceanic tide and the drainage conditions of the trunk line.

**Table 8.** Flood analysis in the Dowon watershed according to the flood damage reduction facility installation conditions.

| Structural Improvement Plan | Number of Inundated Cells | Inundation Time (min) | Inundation Area (m²) | Inundation Volume (m³) | Flood Reduction (%) |
| --- | --- | --- | --- | --- | --- |
| Not installed | 2213 | 580 | 221,207 | 24,028 | - |
| Conduit improvement | 0 | 0 | 0 | 0 | 100.00 |
| Installation of the trunk drainage sewer | 762 | 370 | 76,211 | 10,556 | 65.55 |

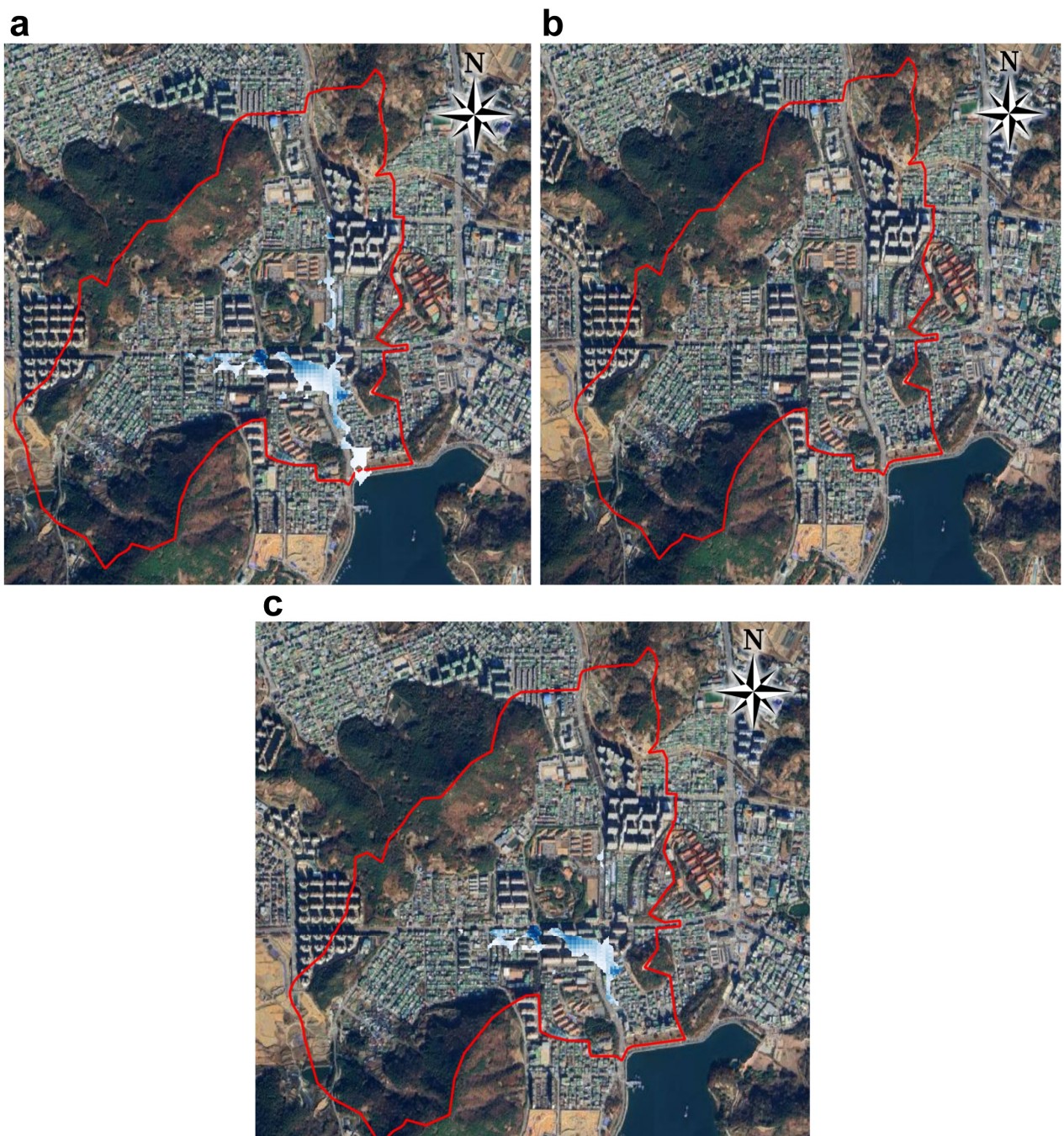

**Figure 16.** Flood analysis in the Dowon watershed according to the flood damage reduction facility installation conditions. (**a**) Not installed, (**b**) conduit improvement, and (**c**) installation for the trunk drainage system.

While the conduit improvement is more effective as a countermeasure against flooding than the installation of the trunk drainage sewer, its estimated total project cost is approximately KRW 24.3 billion. The applicability of the trunk drainage sewer is economically advantageous, with the project cost for installing an approximately 10,000 m$^3$ capacity storage facility being KRW 4.3 billion. As mentioned in the installation conditions and application limitations of the trunk drain washer, the installation of a trunk drain washer alone cannot completely solve the undulation, and there is a limitation in that the effect is limited in watersheds with many branches.

## 4. Conclusions

In this study, the applicability of the trunk drainage sewer system, which is a storage facility linking to the existing rainwater conduit for the smooth exclusion discharge of domestic water in urban areas where rainfall runoff is increasing even with the same intensity of rainfall, was analyzed in three areas with a history of flooding. First, the model was calibrated and verified based on the data of the flood damage history and the amount of runoff. Then, the applicability of the trunk drainage sewer system was analyzed by comparing the number of inundated cells, inundation area, and inundation volume to determine its effectiveness for reducing inundation against the reduction measures taken by the local governments. The main research results are summarized as follows.

1. As a result of calibrating and estimating the SWMM input data based on the actual runoff data for three watersheds and qualitatively evaluating them using the LSSI method, the applicability of the trunk drainage sewer system was "Excellent" in the Gunja watershed and "Good" in the Dorim watershed.
2. The analysis results for each condition of the capacity, location, and discharge point, which are the installation conditions of the trunk drainage sewer, indicated that the peak flow reduction was the greatest at 40% DUAR for a capacity of 1000–5000 m$^3$ and at 60% DUAR for a capacity of 10,000 m$^3$, suggesting that a trunk drainage sewer of greater capacity should be located more downstream to achieve a greater peak flow reduction. However, the influence of the local drainage structure, such as the distribution of branch lines and the presence or absence of drainage pumping stations, should be considered.
3. A comparative analysis of the adequacy of the trunk drainage sewer system as compared to the reduction facility installation project typically planned and implemented in the regions suggested that the hydrological reduction effect following the installation of a trunk drainage sewer of relatively small capacity was significant in terms of the number of inundated cells, inundation time, inundation area, and inundation volume.
4. The trunk drainage sewer system, allowing various combinations of installation conditions, such as location and discharge point, appears to have high applicability in terms of urban planning and economics.

**Funding:** This research received no external funding.

**Data Availability Statement:** No new data were created or analyzed in this study. Data sharing is not applicable to this article.

**Conflicts of Interest:** The author declares no conflict of interest. The funder had no role in the design of the study; in the collection, analyses, or interpretation of data; in the writing of the manuscript; or in the decision to publish the results.

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
