# Peer review of "Applicability Analysis of Trunk Drainage Sewer System for Reduction of Inundation in Urban Dense Areas"

_water, doi:10.3390/w14213399_

Round 1
Reviewer 1 Report
Changjae Kwak et al. analysed a possible use of the trunk drainage sewers to mitigate flood damage in densely populated cities and compared it with the standard abatement procedures. The problem is important in view of climate changes and resulting unexpected weather phenomena. Authors used meteorological records for model simulations. The study was well planned and realised accordingly. Finally, the Authors presented convincing evidence of possible advantages of the trunk drainage sewers (in terms of flood mitigation and of economic costs) over the standard technologies. I have only some minor doubts and comments to the text listed below:
1 – I miss the “temporal” factor in analysing the performance of drainage systems. Although presented in some graphs, the duration of heavy rains and its effect on water discharge capacity is not sufficiently commented.
2 – Graphs and photos presented in figures have poor resolution (at least in my printed copy of the manuscript) and sometimes are hardly visible. I hope these problems will disappear in final publication.
3 – lines 60 – 61. Sea level rise is predicted in every climate change scenario to the year 2100. I doubt, however, that it is an important factor for water discharge on a shorter time scale of years or decades. Instead, I would add the ratio of permeable to impervious areas in large cities as a factor in flood mitigation strategies.
4 – I would suggest careful checking dots and commas in numerical values (see line 283, Table 5).
5 – The number of inundated cells given in line 408 does not correspond to that shown in Table 6).
Reviewer 2 Report
It is not established what the novel or innovative methodology is, which differs from other known results for assessing the applicability of similar systems described in previous research. Better examples of similar research that is not just applying XP SWMM in different areas are, for instance, DOI:10.2166/ws.2017.184 and DOI:10.2166/ws.2008.029, to name Korean authors. It is possible to find documentation of similar analyses at the professional level, including in European projects such as RESCCUE or the US EPA Stormwater Best Management Practice Design Guide.
The only innovation I could notice in the study is the incorporation of the Lee Sallee shape index (LSSI) technique, but it is not explained how it has been applied to the data obtained or the method.
It is recommended that the authors show details of the novel methodologies they have surely applied beyond using XP SWMM and its normal parameters in any engineering project involving a trunk drainage sewer system. The applicability assessment, as described in the article, is an engineering project and does not meet the requirements of a scientific publication. There are no hypotheses or points of comparison from new versus proven methodologies, and the knowledge gap is not fully identified, so the manuscript's scientific content is minimal. The experimental design is inappropriate as it does not differ from an engineering project and does not contribute to filling a knowledge gap.
The Figures do not present sufficient resolution to be understood.
The manuscript could be fascinating as it builds on current data and infrastructure, but applying or generating new techniques to perform such analysis is strongly recommended.
It is unclear what the difference is between a "trunk drainage sewer system drainage method" you propose to analyse its applicability and the ones that have been in place for more than 70 years around the world. Figure 1 and its components are not precise. What is the difference between this type of infrastructure and what currently exists, and is it professionally proven?
